# Hidden Cardiotoxicity of Rofecoxib Can be Revealed in Experimental Models of Ischemia/Reperfusion

**DOI:** 10.3390/cells9030551

**Published:** 2020-02-26

**Authors:** Gábor B. Brenner, András Makkos, Csilla Terézia Nagy, Zsófia Onódi, Nabil V. Sayour, Tamás G. Gergely, Bernadett Kiss, Anikó Görbe, Éva Sághy, Zoltán S. Zádori, Bernadette Lázár, Tamás Baranyai, Richárd S. Varga, Zoltán Husti, András Varró, László Tóthfalusi, Rainer Schulz, István Baczkó, Zoltán Giricz, Péter Ferdinandy

**Affiliations:** 1Department of Pharmacology and Pharmacotherapy, Semmelweis University, 1089 Budapest, Hungary; brenner.gabor@med.semmelweis-univ.hu (G.B.B.); makkos.andras@med.semmelweis-univ.hu (A.M.); nagy.csilla@med.semmelweis-univ.hu (C.T.N.); onodi.zsofia@med.semmelweis-univ.hu (Z.O.); sayour.nabil@gmail.com (N.V.S.); tamas.gergely95@gmail.com (T.G.G.); kiss.bernadett@med.semmelweis-univ.hu (B.K.); gorbe.aniko@med.semmelweis-univ.hu (A.G.); saghy.eva@med.semmelweis-univ.hu (É.S.); zadori.zoltan@med.semmelweis-univ.hu (Z.S.Z.); lazar.bernadette@med.semmelweis-univ.hu (B.L.); baranyai.tamas@med.semmelweis-univ.hu (T.B.); 2Pharmahungary Group, 6722 Szeged, Hungary; 3Department of Pharmacology and Pharmacotherapy, University of Szeged, 6721 Szeged, Hungary; varga.richard@med.u-szeged.hu (R.S.V.); husti.zoltan@med.u-szeged.hu (Z.H.); varro.andras@med.u-szeged.hu (A.V.); baczko.istvan@med.u-szeged.hu (I.B.); 4Department of Pharmacodynamics, Semmelweis University, 1089 Budapest, Hungary; tothfalusi.laszlo@pharma.semmelweis-univ.hu; 5Department of Physiology, Justus-Liebig University, 35392 Giessen, Germany; Rainer.Schulz@physiologie.med.uni-giessen.de

**Keywords:** cardiotoxicity, COX-2, electrophysiology, safety testing, arrhythmia, Vioxx, ischemic conditioning, reperfusion injury, hiddentox, pharmacovigilance

## Abstract

Cardiac adverse effects are among the leading causes of the discontinuation of clinical trials and the withdrawal of drugs from the market. The novel concept of ‘hidden cardiotoxicity’ is defined as cardiotoxicity of a drug that manifests in the diseased (e.g., ischemic/reperfused), but not in the healthy heart or as a drug-induced deterioration of cardiac stress adaptation (e.g., ischemic conditioning). Here, we aimed to test if the cardiotoxicity of a selective COX-2 inhibitor rofecoxib that was revealed during its clinical use, i.e., increased occurrence of proarrhythmic and thrombotic events, could have been revealed in early phases of drug development by using preclinical models of ischemia/reperfusion (I/R) injury. Rats that were treated with rofecoxib or vehicle for four weeks were subjected to 30 min. coronary artery occlusion and 120 min. reperfusion with or without cardioprotection that is induced by ischemic preconditioning (IPC). Rofecoxib increased overall the arrhythmias including ventricular fibrillation (VF) during I/R. The proarrhythmic effect of rofecoxib during I/R was not observed in the IPC group. Rofecoxib prolonged the action potential duration (APD) in isolated papillary muscles, which was not seen in the simulated IPC group. Interestingly, while showing hidden cardiotoxicity manifested as a proarrhythmic effect during I/R, rofecoxib decreased the infarct size and increased the survival of adult rat cardiac myocytes that were subjected to simulated I/R injury. This is the first demonstration that rofecoxib increased acute mortality due to its proarrhythmic effect via increased APD during I/R. Rofecoxib did not interfere with the cardiprotective effect of IPC; moreover, IPC was able to protect against rofecoxib-induced hidden cardiotoxicity. These results show that cardiac safety testing with simple preclinical models of I/R injury uncovers hidden cardiotoxicity of rofecoxib and might reveal the hidden cardiotoxicity of other drugs.

## 1. Introduction

Unexpected clinical cardiotoxicity is still the leading cause of discontinuation of clinical trials and the withdrawal of drugs from the market despite great efforts to detect cardiotoxicity in the preclinical phase of drug development programs [1]. Such cardiotoxic effects remain undetected during preclinical and early clinical safety studies and they may manifest in the presence of cardiac diseases e.g., in myocardial I/R conditions; therefore, we termed this phenomenon “hidden cardiotoxicity” [2].

Hidden cardiotoxicity often manifests as ischemia-related lethal myocardial injury and/or as I/R-induced arrhythmias and/or as cardiac dysfunction. Thus, drugs with hidden cardiotoxic properties may present as a serious risk to patients as drugs with overt cardiotoxicity, such as certain cancer treatments [3]. The mechanisms of hidden cardiotoxicity may include the activation of cell death- or pro-arrhythmic processes during cardiac I/R, as well as the inhibition of cardioprotective signaling pathways (e.g., ischemic conditioning-induced protection), either of which may be aggravated by the presence of cardiovascular comorbidities [2]. The mechanism of cardiovascular toxicity of cancer treatments is described elsewhere in detail.

Nearly 500 medicinal products were withdrawn from the market between 1953 and 2013 [4], the majority of which is related to cardiac adverse events [1]. Moreover, an estimated 197,000 deaths are attributed to adverse drug reactions in the European Union each year [5].

Hidden cardiotoxicity remains undetected in the preclinical and early clinical phases of drug development, since the current guidelines only require the assessment of drug safety in healthy animals [6,7]. In addition, preclinical and clinical cardiac electrophysiological safety test guidelines advocate the use of healthy animals, tissues, and healthy human volunteers for the assessment of the pro-arrhythmic adverse effects of compounds in development and these tests do not represent patients with increased arrhythmia susceptibility [8,9]. However, in clinical trials, cardiotoxic adverse events occur in an unpredictable manner, often in patients with cardiac diseases and/or with cardiovascular comorbidities, e.g., hyperlipidemia, hyperglycemia, hypertension, aging, or inflammatory diseases [10,11]. Indeed, the guidelines for the treatment of heart failure by the American College of Cardiology Foundation/American Heart Association recommend avoiding the use of certain medications in heart failure, e.g., cyclooxygenase-2 (COX-2) inhibitors, since they may exacerbate underlying myocardial dysfunction [12].

Rofecoxib, which is a COX-2 inhibitor, was withdrawn from the market due to an increased risk of cardiovascular prothrombotic events being observed in the VIGOR and APPROVe trials [13,14]. Later, in a meta-analysis that included 116,094 participants, it was shown that the use of rofecoxib was associated with an increased risk of arrhythmias [15]. Several other mechanism have been proposed for rofecoxib-induced cardiotoxicity, such as the inhibition of protection against I/R injury, prevention of production of epi-lipoxins, increase in blood pressure, and inhibition of vascular remodeling, however, none of those has been detected during preclinical safety assessment [16]. Nevertheless, according to our definition, rofecoxib showed hidden cardiotoxic properties; earlier and appropriate preclinical tests could have revealed these effects, thus preventing a number of serious adverse events, thereby increasing patient safety. The fact that the cardiotoxic effects of rofecoxib remained hidden in preclinical studies and was only revealed in phase 4 clinical studies and by a following metaanalysis and the enormous costs of long-term cardiovascular outcome trials required to reveal hidden cardiotoxicity suggest that more sensitive screening methods are required for toxicity studies including animal models of myocardial I/R and/or comorbidities [17,18].

To this end, here we aimed to investigate that hidden cardiotoxicity of rofecoxib, that remained unrevealed during preclinical safety assessment, could have been detected before its authorization in pathological conditions while using cellular-, isolated tissue-, and small animal models of acute I/R injury, and to test the effect of rofecoxib on cardioprotection that is elicited by IPC. Our results show that hidden cardiotoxic property of rofecoxib can be revealed with preclinical models of I/R injury. Safety testing of other drugs in the presence of I/R might uncover their hidden cardiotoxicities.

## 2. Materials and Methods

### 2.1. Ethical Considerations

This investigation complies with the Guide for the Care and Use of Laboratory Animals published by the US National Institutes of Health (NIH publication No. 85–23, revised 1996), to the EU Directive (2010/63/EU). The current study is based on the rule of the replacement, refinement, or reduction (3Rs). The animal ethics committee of the Semmelweis University, Budapest, Hungary, and by the National Scientific Ethical Committee on Animal Experimentation approved it, and it was permitted by the government (Food Chain Safety and Animal Health Directorate of the Government Office for Pest County (PE/EA/1784-7/2017) and by the Ethical Committee for the Protection of Animals in Research of the University of Szeged, Szeged, Hungary (I-74-24-2017), and by the Department of Public Health and Food Control of the Government Office for Csongrád County, Hungary (XIII/3331/2017).

### 2.2. Sources of Chemicals

Most of the chemicals were purchased from Sigma (St. Louis, MO, US), including hydroxyethylcellulose (#09368), Evans blue dye (#E2129), triphenyltetrazolium chloride (#T8877), HEPES buffer (#H3375), dimethyl-sulfoxide (DMSO #D5879 or #D2650), 2-Deoxy-D-glucose (#D8375), and laminin (#L2020). Other sources: heparin (Merck, Darmstadt, Germany, #375095), rofecoxib (MedChemExpress Europe, Sollentuna, Sweden, #HY-17372), pentobarbital (Produlab Pharma, Raamsdonksweer, The Netherlands, #17F015), MgSO_4_ (Reanal, Budapest, Hungary, #20341), collagenase II (Biochrom GmbH, Berlin, Germany, #c2-22), fetal bovine serum (FBS, EuroClone, Pero MI, Italy, #ECS0180L), M199 (Lonza, Verviers, Belgium, #BE12-117F), Bovine Serum Albumin (BSA, Santa Cruz Biotechnology, Santa Cruz CA, USA, #sc-2323), calcein AM (PromoCell GmbH, Heidelberg, Germany, #PK-CA707-80011-3), Dulbecco’s Phosphate-Buffered Saline (DPBS, Gibco, Grand Island New York, USA, #14080-055).

### 2.3. In Vivo Ischemia/Reperfusion Injury Study

For *in vivo* experiments, male Wistar rats of 187–287 g were treated with 5.12 mg kg^−1^ rofecoxib or with its vehicle, 1% hydroxyethylcellulose by oral gavage once daily for 28 ± 1 days. The dose of rofecoxib was extrapolated from the daily human dose (50 mg daily) that showed cardiovascular side effects in clinical studies [13] by using the formula that was described by Reagan-Shaw *et al.* [19]:(1)$$\mathrm{HED}\;\left(\mathrm{mg}\times {\mathrm{kg}}^{-1}\right)=\mathrm{Animal}\;\mathrm{dose}\;\left(\mathrm{mg}\times {\mathrm{kg}}^{-1}\right)\times \frac{\mathrm{rat}\;K_{m}}{\mathrm{human}\;K_{m}}$$

Animal dose was calculated by dividing the HED (human equivalent dose, 50 mg 60 kg^−1^ for average 60 kg adult) by the ratio of average rat correction factor (rat K_m_ = 6) and the average human correction factor (human K_m_ = 37).

The animals were housed in a temperature (22 ± 2 °C)-, and humidity-controlled room at a 12 h light/dark cycle and had free access to laboratory chow and drinking water *ad libitum*. Groups of animals treated with rofecoxib or vehicle for 28 ± 1 days and weighing 320–440 g were anaesthetized with 60 mg kg^−1^ pentobarbital intraperitoneally. The absence of pedal reflex was considered as being deep surgical anesthesia. Body surface electrocardiogram (ECG) was monitored throughout the experiments by using standard limb leads (AD Instruments, Bella Vista, Australia). Cannulated right carotid artery was used for the measurement of mean arterial blood pressure (MAP, AD Instruments, Bella Vista, Australia) and fluid supplementation with saline containing 10 IU kg^−1^ heparin. The core body temperature was maintained at physiological temperature with a heating pad (Harvard Apparatus, Holliston, Massachusetts). Rats were ventilated with rodent ventilator (Ugo-Basile, Gemonio, Italy) with room air in a volume of 6.2 mL kg^−1^ and frequency of 69 ± 3 breath min.^−1^. A 5-0 Prolene suture (Ethicon, Johnson & Johnson, Budapest, Hungary) was looped around the left anterior descending (LAD) coronary artery [20]. After the previously listed procedures the 0 min. of experiment was appointed. The animals received intraperitoneal injection of 100 IU kg^−1^ heparin at 35, 65, and 185 min. of experiments.

Experimental design and study protocols are illustrated in Figure 1. Altogether 62 animals were included in the *in vivo* experiments. In order to achieve comparable number of surviving animals in each group, based on our preliminary observations 30% more animals were assigned to the rofecoxib-treated group (*n* = 35) than to the vehicle-treated group (*n* = 27). Rofecoxib- and vehicle-treated animals were then subjected to I/R with or without IPC using directed randomization during the study to assign more animals to the higher mortality groups: I/R+vehicle group (*n* = 11), I/R + rofecoxib group (*n* = 18), IPC+vehicle group (*n* = 16) and IPC+rofecoxib group (*n* = 17). I/R was induced by 30 min. LAD occlusion and IPC was elicited by 3 cycles of brief 5-min. LAD occlusion and 5 min. reperfusion before I/R. Animals received a 120 min. reperfusion. Appearance of ischemia was confirmed by ST segment elevation or depression, appearance of arrhythmias and pallor of the myocardial regions distal to the site of occlusion.

#### 2.3.1. Mortality Analysis

The cause of death was classified as either irreversible VF, pulseless electrical activity, and bradycardia (<150 BPM), accompanied by hypotension (MAP < 15 mmHg).

#### 2.3.2. Arrhythmia Analysis

The incidence and duration of arrhythmias occurring during 30 min. of ischemia and the first 15 min. of reperfusion were analyzed by two investigators independently in a blinded fashion according to the Lambeth conventions and quantified while using the “score A” described by Curtis and Walker [21,22]. The 45 min-long ECG records were divided into five-minute periods, and then each interval was scored according to most severe arrhythmia type in the given interval. In the case of fatal VF, scores were kept throughout the subsequent periods. The arrhythmia maps were constructed by using a color scale, where the 5-min. periods were colored according to the most severe arrhythmia type.

#### 2.3.3. Infarct Size Measurement

After 120 min. of reperfusion hearts were excised and perfused for 2 min. with oxygenated Krebs-Henseleit solution (in mM: NaCl 118, KCl 4.7, MgSO_4_ 1.2, CaCl_2_ 1.25, KH_2_PO_4_ 1.2, NaHCO_3_ 25, and glucose 11) at 37 °C in Langendorff mode to remove blood from the tissue, LAD was re-occluded, and the area at risk (AAR) was negatively stained with Evans blue dye through the ascending aorta. For the assessment of viable myocardial tissue, 2 mm-thick slices were cut and incubated in 1% triphenyltetrazolium chloride at 37 °C for 14 min. The slices were weighed and scanned. Planimetric analyses were performed by two independent, blinded investigators with InfarctSize 2.4b software (Pharmahungary Group, Budapest, Hungary). Area at risk (AAR) was expressed as the proportion of the left ventricular area, and the infarct size as the proportion of the AAR, and then areas were normalized to the mass of each slice.

### 2.4. Ex Vivo Simulated Ischemia/Reperfusion Injury Study

For isolated papillary muscle experiments, male Wistar rats weighing 200–250 g were used. The rats were anesthetized with pentobarbital intraperitoneally (30 mg kg^−1^), followed by rapid excision of the heart via thoracotomy. Left ventricular papillary muscle preparations were mounted in a tissue chamber (volume together with solution reservoir: 50 mL) and they were then continuously perfused with oxygen–saturated, HEPES-buffered Tyrode’s solution (in mM: NaCl 144, NaH_2_PO_4_ 0.4, KCl 4, MgSO_4_ 0.53, CaCl_2_ 1.8, glucose 5.5, HEPES 5 at pH 7.4, 37 °C). The preparations were stimulated (Hugo Sachs Electronik stimulator type 215/II, March-Hugstetten, Germany) at a cycle length of 1000 ms (frequency: 1 Hz), while using 2 ms-long rectangular constant voltage pulses that were isolated from ground and delivered across bipolar platinum electrodes in contact with the preparation. Transmembrane potentials were recorded while using the conventional microelectrode technique. Microelectrodes that were filled with 3 M KCl and exhibiting tip resistances of 5–20 MΩ were connected to a high impedance electrometer (type 309, MDE Heidelberg GMBH, Heidelberg, Germany) coupled to a dual beam oscilloscope (Tektronix, Beaverton, OR, USA).

Figure 2 illustrates the experimental design and study protocols. Altogether, 54 animals were included in the *ex vivo* experiments. Papillary muscles of 6 animals/group were superfused with oxygen–saturated HEPES-buffered Tyrode’s solution (normoxic solution) and were allowed to equilibrate for 60 min. before baseline measurements were taken. Throughout the experiments, measurements were taken every 2 min. Following the 60-min. equilibration period, groups of preparations were superfused with normoxic solution containing either vehicle, 1 or 10 µM rofecoxib (Normoxia groups) dissolved in DMSO for 90 min. The concentration of 1 µM was chosen for rofecoxib based on the peak plasma concentration (C_max_) measured after a single, 5 mg kg^−1^ oral dose of rofecoxib in rats [23]. The highest final concentration of DMSO following the application of 10 µM rofecoxib was 0.2% in the solution. Following the 60-min. baseline superfusion, groups of preparations were superfused with normoxic solution for 30 min. then with nitrogen-saturated and HEPES-buffered solution (ischemic solution, in mM: NaCl 144, NaH_2_PO_4_ 0.4, KCl 4, MgSO_4_ 0.53, CaCl_2_ 1.8, 2-deoxy-D-glucose 5.5, HEPES 5 at pH 6.9, and 37 °C) for 30 min., and then with normoxic solution for 30 min., all containing either vehicle, 1 or 10 µM rofecoxib (sI/R groups) to induce simulated I/R (sI/R). In additional groups of preparations, sIPC (sIPC groups) was performed before 30 min. ischemia by using the following protocol: three times 5-min. simulated ischemia with intermittent 5 min. reperfusion periods. Before index ischemia, the last reperfusion lasted 15 min.

#### Evaluation of Action Potential Parameters

Unbiased evaluation of action potential parameters was achieved by automatic evaluation while using software that was developed in Department of Pharmacology and Pharmacotherapy, University of Szeged (Hugo Sachs Electronic-Action Potential Evaluation System): V_max_, CT, RMP, APA, APD at 75 and 90% of repolarization (APD_75_ and APD_90_, respectively). The maintenance of the same impalement throughout each experiment was attempted. However, in case an impalement was dislodged, electrode adjustment was performed, and the experiment was terminated and all data were excluded from analysis if the action potential characteristics of the re-established impalement deviated by more than 5% from the previous measurement.

### 2.5. In Vitro Simulated Ischemia/Reperfusion Injury Study

For *in vitro* cell culture experiments, male Wistar rats weighing 150–200 g were used. The rats were anesthetized with pentobarbital intraperitoneally (60 mg kg^−1^) and each animal was heparinized (500 IU/kg) through femoral vein. The hearts were excised, cannulated, and perfused retrograde with Krebs–Henseleit solution to wash out the blood. Then hearts were perfused with collagenase II (8000 U/mL) containing Krebs solution for 30–45 min. Subsequently, the ventricles were removed and then chopped in small pieces and digestion continued for more 10 min. The cell suspension was filtrated and pelleted under gravity, repeated 2–3 times. Under these steps, the Ca^2+^ concentration was gradually increased up to a final of 1 mM. The isolated cells (7500 cell/well) were plated in laminin-coated wells of a 24-well plate (Thermo Fisher Scientific, Waltham, USA) and incubated for 3 h in proliferation media (5% FBS containing M199) and in growth media (serum free M199) for 24 h. The experimental design and study protocols are illustrated on Figure 3. Altogether 12 animals were included in the *in vitro* experiments. To achieve the *n* = 6 group size in all the different normoxic and sI/R groups, i.e., six separate series of cell isolation procedures were made using two hearts, one for the normoxic and one for the sI/R groups. After 24 h, growth media was replaced with growth media containing vehicle or rofecoxib in increasing doses (0.1, 0.3, 1, 3, and 10 µM) and the cells were kept in CO_2_ incubator (Scancell - Labogene, Lynge, Denmark) for 60 min. [24]. After 60 min. in groups of cells, growth media was replaced for 180 min. with either normoxic solution (in mM: NaCl 125, KCl 5.4, NaH_2_PO_4_ 1.2, MgCl_2_ 0.5, HEPES 20, MgSO_4,_ 1.3, CaCl_2_ 1, glucose 15, taurine 5, creatine-monohydrate 2.5 and BSA 0.1%, pH 7.4) in CO_2_ incubator (Normoxia groups) or with hypoxic solution (in mM: NaCl 119, KCl 5.4, MgSO_4_ 1.3, NaH_2_PO_4_ 1.2, HEPES 5, MgCl_2_ 0.5, CaCl_2_ 0.9, Na-lactate 20, BSA 0.1% pH 6.4) in a three-gas (95% N_2_ and 5% CO_2_) incubator (sI/R groups, Panasonic Healthcare Co., Ltd., Gunma, Japan), both containing the fore-mentioned doses of rofecoxib or vehicle only [25]. Following the 180 min. normoxic or sI/R conditions cells were kept in growth medium containing vehicle or rofecoxib in increasing doses in CO_2_ incubator.

#### Viability Assay

Calcein staining was performed to assess cell viability [26]. The cells were washed with warm DPBS and calcein solution (1 µM) was added and incubated for 30 min. at room temperature in a dark chamber. Afterwards, the calcein solution was replaced with fresh DPBS. An unbiased evaluation was performed by automatic detection of the fluorescence intensity of each well by Varioskan Lux multimode microplate reader (Thermo Fisher Scientific, Waltham, USA) at temperature: 37 °C; excitation wavelength: 490nm; emission wavelength: 520 nm. Autofluorescence of rofecoxib that was measured in DPBS (0.1, 0.3, 1, 3 and 10 µM) was not detected; therefore, interference did not influence the results of the viability assay. Six separate technical repeats were performed, and an average of four wells/group/repeats are presented on the graph. The cell survival data are expressed as relative fluorescence units (RFU). Normoxia + vehicle group was set to 1 RFU arbitrary unit and all data were normalized to the averaged sI/R group.

### 2.6. Statistical Analysis

The Odds Ratio (OR) with 95% confidence interval (CI 95%) was estimated by logistic regression to identify the mortality rate differences among the treatment groups. Continuous data are shown as mean ± standard error. The difference between treatment groups was evaluated while using two-way ANOVA or one-way ANOVA followed by Fisher LSD post hoc tests with multiple comparisons (for in vivo infarct size analysis, for ex vivo action potential measurements, and for in vitro cell viability study) and two-way repeated measures ANOVA, followed by Fisher LSD post hoc test (for in vivo arrhythmia analysis). We used GraphPad Prism (version 6.0, GraphPad Software, California, USA) and R (version 3.4) with the lme4 library. We claimed that the differences were statistically significant if *p* < 0.05.

## 3. Results

### 3.1. Chronic Rofecoxib Treatment Increased Acute Mortality During Cardiac Ischemia/Reperfusion

The rats were treated with rofecoxib for four weeks and then subjected to 30 min. ischemia and 120 min. reperfusion to investigate hidden cardiotoxicity of rofecoxib. Rofecoxib treatment increased the mortality rate as compared to the pooled data of other groups (OR = 7.73, CI 95% = 1.70–34.97 vs. I/R + vehicle + IPC+vehicle + IPC+rofecoxib; *p* < 0.008; Figure 4). In the I/R+rofecoxib group, seven animals died due to irreversible VF during the ischemic period and one animal died due to a sudden drop in blood pressure during reperfusion. In the I/R + vehicle group, only one animal died due to irreversible VF during the ischemic period. Animals died during the short I/R stimuli of IPC (six/each IPC group) were excluded from further evaluations and are not shown in Figure 3, Figure 4 and Figure 5. In the IPC + rofecoxib group, one animal died due sudden drop in blood pressure in the reperfusion period.

### 3.2. Chronic Rofecoxib Treatment Increased Arrhythmia Score in Cardiac Ischemia/Reperfusion

The severity and duration of arrhythmias were evaluated by scoring 5 min. intervals according to the Lambeth conventions during cardiac ischemia and early reperfusion. The results are represented as an arrhythmia map in Figure 5, showing the type of most severe arrhythmias occurring during a given 5-min. interval by a color scale.

The peak arrhythmia scores were achieved in the I/R + vehicle groups after 10 min. of ischemia (50^th^ min. of experiment) and, following that, they rapidly decreased (Figure 6). In contrast, in the I/R + rofecoxib group the initial increase runs parallel with the I/R + vehicle group, but, following that, the decline is much slower. We tested the statistical hypothesis that scores decrease in parallel by fitting the linear mixed regression model on the observations by excluding the data of the first period. The difference between slopes was highly significant (*p*
_Time x Group interaction_ = 0.00681), thus suggesting a pronounced effect of rofecoxib on the recovery. Yet, such a difference does not exist between the estimated peak values at the end of 50^th^ minute of experiment (*p* = 0.66367). The initial increase of arrhythmia scores was not observed in the IPC groups.

### 3.3. Rofecoxib Decreased Infarct Size and Did Not Interfere with Cardioprotection by Ischemic Preconditioning

We measured infarct size to explore the effect of rofecoxib on I/R injury and cardioprotection by IPC. Rofecoxib reduced infarct size (I/R + rofecoxib) as compared to the vehicle-treated (I/R + vehicle) group (Figure 7). Infarct size was significantly smaller in the IPC+vehicle group as compared to I/R+vehicle. Chronic rofecoxib treatment did not affect infarct size-limiting effect of IPC in IPC+rofecoxib when compared to the IPC+vehicle group.

No significant difference was observed between groups for the AAR expressed as a percentage of the left ventricle (I/R + vehicle: 51.1 ± 2.8%; IPC + vehicle: 41.6 ± 2.3%; I/R + rofecoxib: 44.8 ± 4.1%; IPC+rofecoxib 50.6 ± 4.9%).

### 3.4. Rofecoxib Increased the Action Potential Duration in Rat Isolated Papillary Muscles at the End of Simulated Ischemia/Reperfusion and this Effect Was Not Observed Ischemic Preconditioning Groups

*In vitro* simulated ischemia/reperfusion (sI/R) and sIPC experiments were performed on isolated rat left ventricular papillary muscles in order to analyze the effect of rofecoxib on cardiac action potential parameters. Rofecoxib treatment did not change any of the investigated electrophysiological parameters, including APD_90_ (Figure 8A,B) and APD_75_ (Appendix A) in normoxic conditions. As expected, the 30 min. simulated ischemia significantly shortened APD_90_ (Figure 8A) and APD_75_ (Appendix A) in all groups that were subjected to ischemia when compared to the respective normoxic groups. However, importantly, in the presence of sI/R rofecoxib dose-dependently increased APD_90_ (Figure 8B) and increased APD_75_ (Appendix A) upon reperfusion following the 30 min. simulated ischemia. In the sIPC group, these effects of rofecoxib on APD were not seen during reperfusion (Figure 8B and Appendix A). The effects of rofecoxib on action potential amplitude (APA), conduction time (CT), resting membrane potential (RMP), and maximum rate of depolarization (V_max_) in sI/R are detailed in Appendix A. Simulated ischemia (30 min.) resulted in an increase of CT in all groups, while V_max_ was significantly reduced in the rofecoxib treated groups (Appendix A), possibly indicating decreased sodium channel function following rofecoxib administration in ischemic conditions only.

### 3.5. Rofecoxib Treatment Increased Viability of Isolated Adult Rat Cardiac Myocytes in Normoxia and in Simulated Ischemia/Reperfusion Injury

*In vitro* sI/R experiments were performed in order to analyze the effect of rofecoxib on viability of isolated cardiac myocytes. sI/R caused significant cell death (Figure 9) as compared to normoxic control, which was reversed by rofecoxib treatment at 0.1, 0.3, 1, and 3 µM concentration, respectively, thereby supporting the *in vivo* data showing the infarct size reduction by rofecoxib (Figure 7).

## 4. Discussion

Here, we demonstrated, for the first time, in the literature that rofecoxib increased acute mortality due to its proarrhythmic effect via increased APD during I/R. We also showed that rofecoxib did not interfere with the cardiprotective effect of IPC and that IPC was able to protect against rofecoxib-induced hidden cardiotoxicity.

In the present study, we have shown that chronic rofecoxib treatment increased mortality after acute cardiac I/R in rats. Increased mortality due to rofecoxib have also been reported in clinical trials. Gislason *et al.* concluded that selective COX-2 inhibition with rofecoxib and celecoxib increased the mortality at all doses in patients with prior myocardial infarction; however, the underlying mechanisms were not studied [27]. Our present data imply that the increased mortality due to rofecoxib treatment can be attributed to its proarrhythmic property that only manifests following I/R. Myocardial ischemia *per se* renders myocardial tissue more susceptible to ventricular arrhythmias and I/R injury may exacerbate proarrhythmic effects of drugs [2]. To further analyze the hidden cardiotoxic effects of rofecoxib, we subjected left ventricular papillary muscles to sI/R. Rofecoxib did not change the action potential parameters in normoxic conditions; however, following sI/R, several, potentially proarrhythmic effects appeared. First, the APD was only significantly prolonged by rofecoxib in cardiac tissue that were subjected to sI/R. Increased spatial dispersion of repolarization between normoxic and ischemic myocardium is a critically important factor that promotes the development of ischemia-induced arrhythmias [28,29]. Based on the present results, rofecoxib might further exacerbate the differences in APD between normoxic and ischemic myocardium, further increasing the arrhythmia substrate in I/R. Secondly, simulated ischemia in the presence of rofecoxib more markedly reduced action potential upstroke (characterized by decreased V_max_) by the end of test ischemia, which only suggested an additional reduction of sodium channel function by rofecoxib in sI/R conditions, further decreasing the already slowed impulse conduction in depolarized ischemic myocardial tissue [30].

These data suggest that adverse effects of COX-2 inhibitors may occur only in the presence of cardiac I/R. Our present results are in line with clinical data, as a comprehensive meta-analysis of 114 randomized trials reported increased arrhythmia risks in rofecoxib-treated patients and found that a time-cumulative meta-analytic approach would have revealed its cardiotoxicity earlier [15]. However, a subgroup analysis of patients with ischemic heart diseases was not performed in this meta-analysis, which might have revealed its cardiotoxicity even earlier. These results clearly show that the exclusion of patients with preexisting cardiovascular diseases from clinical studies and the lack of subgroup analyses on patients with underlying co-morbidities in clinical trials may lead to the loss of valuable safety information on drugs with potential hidden cardiotoxic effects [2,31]. Our current preclinical results are in line with previous clinical data. Although the repeated administration of valdecoxib had no effect on QTc interval duration in healthy volunteers [32], and valdecoxib and parecoxib did not increase the risk for cardiac adverse events in patients recovering from major noncardiac surgical procedures [33], but the use of valdecoxib and parecoxib in patients that were subjected to coronary-artery bypass grafting (CABG) was associated with an increased incidence of cardiovascular events [34]. However, in the latter studies, the incidence of arrhythmias was not evaluated. Furthermore, mortality was increased by celecoxib in a chronic post-myocardial infarction-induced heart failure in pigs due to left ventricular rupture and cardiac decompensation [35].

Here we also investigated the interaction of rofecoxib with IPC since the hidden cardiotoxic effect of drugs mightb manifest not only as aggravation of I/R injury, but also as attenuation of ischemic adaptation of the myocardium by ischemic conditioning [2]. We found that rofecoxib alone decreased infarct size and did not interfere with the protective effect of IPC. We tested the cytoprotective effect of rofecoxib in cardiac myocytes subjected to sI/R and found an increased cell survival due to rofecoxib treatment to further test whether the infarct size limiting effect was due to direct cardio-cytoprotective effect or was an artefact due to the significantly less survival of animals with larger infarct size. These results show that rofecoxib has a direct cardio-cytoprotective effect. An infarct size-limiting effect was also shown in rats while using another COX-2 inhibitor celecoxib. Furthermore, in this study mortality rate of celecoxib treatment was not reported, which can also significantly alter the outcomes of cardioprotection [20]. DFU, a compound that is structurally related to rofecoxib, led to a significant improvement in left ventricular end-diastolic pressure and LV systolic pressure and a reduction in infarct size after myocardial infarction in Lewis male rats [36]. A neutral effect on infarct size was shown in different animal models of myocardial infarction with the use of various COX-2 inhibitors [37,38,39,40]. In contrast, Inserte *et al*. showed that in transgenic mice constitutively expressing human COX-2 in cardiomyocytes functional recovery was improved, cell death was reduced after 40 min. of *ex vivo* ischemia, and that pretreatment of mice with the COX-2 inhibitor DFU attenuated cardioprotection [41]. In a recent publication, inducible cardiac-specific COX-2 overexpression showed a infarct-limiting effect in mice [42]. These results suggest that the presence and/or extent of cardioprotection by COX-2 inhibitors may vary due to both the nature of the applied inhibitor and the model species according to different anatomy, physiology, and pharmacokinetics, dosing, etc., and due to differences in surgery protocols (anesthetics, co-medications, chronic, or acute cardiac ischemia etc.). Confirming our finding on direct cardioprotection by rofecoxib *in vivo*, here we also showed that rofecoxib increased cell survival in sI/R on the isolated cardiomyocytes. These findings are in line with a previous report, where the cytoprotective effects of COX-2 inhibition was demonstrated in H9c2 cells and primary rat cardiomyocytes in a simulated hypoxia/reoxygenation (H/R) model, showing that a pretreatment with NS-398 significantly attenuated H/R-induced cellular injury [43]. Therefore, a Janus-like nature of COX-2 inhibition on cardioprotection seems plausible. Here, we found that rofecoxib pretreatment did not affect the infarct size limiting effect of IPC. Confirming our results, parecoxib administered intravenously 15 min. prior to IPC did not affect the infarct size limiting effect of IPC in male Wistar rats [44]. Somewhat in contrast, the protective effect of late phase of ischemic conditioning was attributed to the increased expression and activity of COX-2 [45,46]. Similarly, Sato *et al.* showed an upregulation of COX-2 expression in Harlan Sprague Dawley rats in the ischemic-reperfused cardiac region by late preconditioning, but not by postconditioning and the use of celecoxib completely abrogated the infarct-sparing effect of the combination of two interventions [47]. In summary, it is unlikely that a class effect of COX-2 inhibitors regarding their influence on cardioprotection by ischemic conditioning exist. Moreover, it should be emphasized that, in our present study, the action potential-prolonging effect of rofecoxib in papillary muscle preparations subjected to sI/R was not observed when sIPC was applied. These results show that the hidden cardiotoxic effects of rofecoxibcan be prevented by ischemic conditioning. Similarly, Maulik *et al*. showed that sIPC protected primary adult rat cardiomyocytes against the direct cardiotoxic effect of doxorubicin [48]. However, so far, there are no clinical data on the potential protective effect of IPC on drug-induced cardiotoxicity. The currently ongoing (or still unpublished) ERIC-ONC trial aimed to demonstrate whether remote ischemic preconditioning (RIC) reduces subclinical myocardial injury due to anthracycline chemotherapy [49].

## 5. Conclusions

In conclusion, this is the first demonstration that the hidden cardiotoxicity of rofecoxib can be revealed by preclinical cardiotoxicity testing while using experimental I/R models. Moreover, IPC might protect against the hidden cardiotoxic effects of rofecoxib in vitro. These results show that cardiac safety testing with simple preclinical models of I/R injury uncovers the hidden cardiotoxicity of rofecoxib and might reveal hidden cardiotoxicity of other drugs.

## 6. Limitations

Our goal to show that hidden cardiotoxicity of rofecoxib can be revealed in a preclinical model of I/R injury has been achieved by showing its proarrhythmic properties occurred during its clinical use. A limitation of the study is that we did not assess the molecular mechanism of rofecoxib-induced prolongation of action potential, arrhythmias, and cardioprotection. The proarrhythmic effect of rofecoxib in I/R conditions might be attributed to reduced function of sodium channels; however, this was not evaluated in this study mechanistically.

## Figures and Tables

**Figure 1 cells-09-00551-f001:**
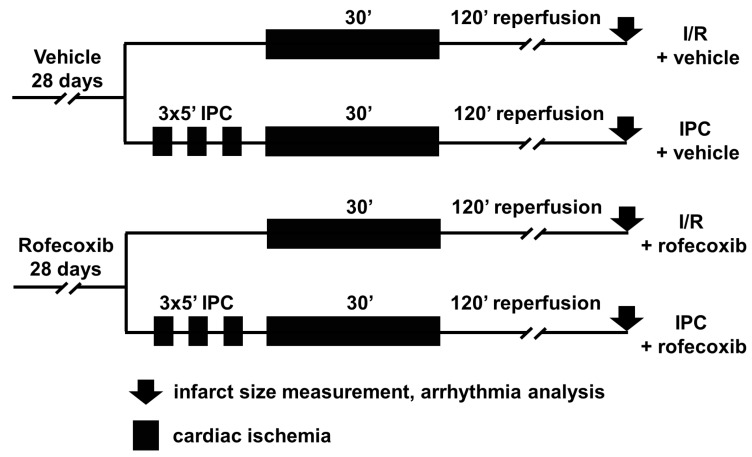
*In vivo* ischemia/reperfusion (I/R) injury study protocol: male Wistar rats treated with rofecoxib (5.12 mg kg^−1^/day) or vehicle for 4 weeks were subjected to I/R of the left anterior descending (LAD) coronary artery or to ischemic preconditioning (IPC) elicited by three cycles of 5 min. LAD occlusion and 5 min. reperfusion before the index ischemia.

**Figure 2 cells-09-00551-f002:**
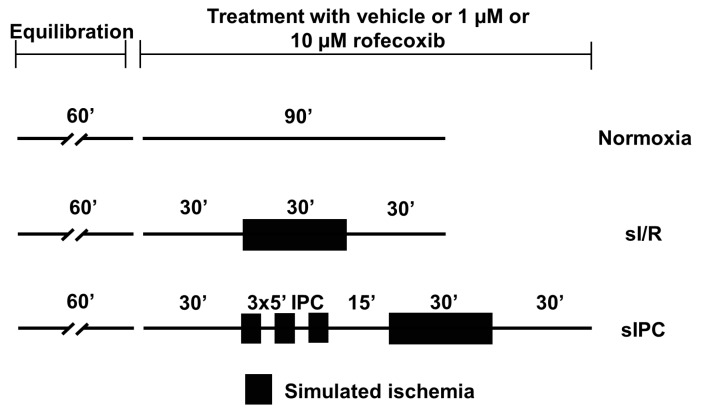
*Ex vivo* simulated ischemia/reperfusion (sI/R) injury study protocol: action potential parameters were measured in isolated rat left ventricular papillary muscles in normoxic, sI/R and simulated ischemic preconditioning (sIPC) conditions in the presence of vehicle or 1 or 10 µM rofecoxib, respectively.

**Figure 3 cells-09-00551-f003:**
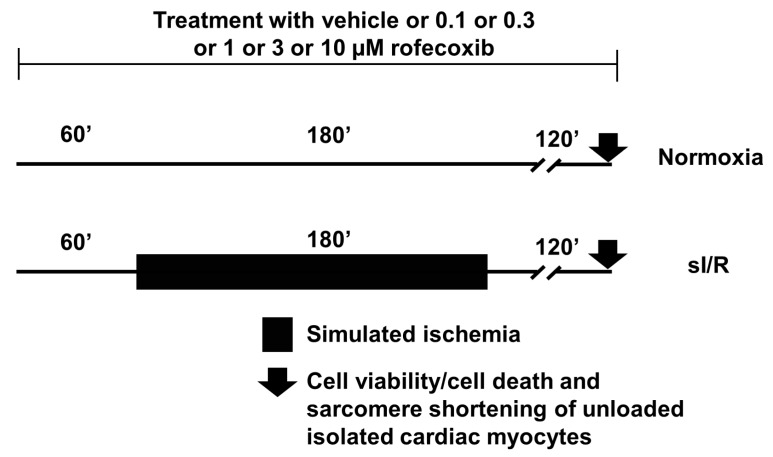
*In vitro* simulated ischemia/reperfusion (sI/R) injury study protocol: cell viability of cultured isolated cardiac myocytes was measured in normoxic and sI/R conditions in the presence of vehicle, 0.1, 0.3, 1, 3, or 10 µM rofecoxib, respectively.

**Figure 4 cells-09-00551-f004:**
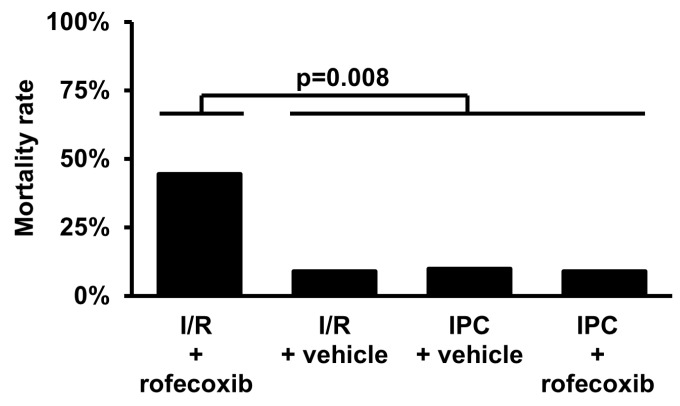
Rofecoxib treatment increased the mortality rate in the ischemia/reperfusion (I/R) group *in vivo*. When compared to the pooled data of other groups, the mortality-increasing effect of rofecoxib was significant (OR = 7.73, CI 95% = 1.70–34.97, *p* < 0.008). IPC: ischemic preconditioning.

**Figure 5 cells-09-00551-f005:**
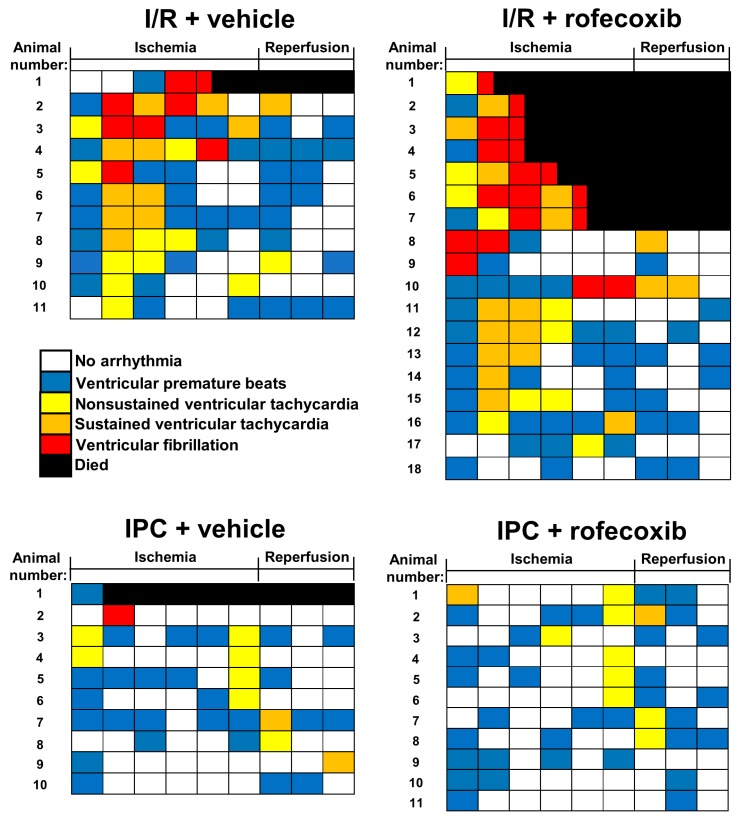
Arrhythmia maps showing the arrhythmias in the order of severity during 30 min. ischemia and at the first 15 min. of reperfusion. Each row represents arrhythmias of each animal. The different color boxes show 5 min. periods. The animals died during the IPC (ischemic preconditioning) are not shown. In the I/R + rofecoxib group animals 1–7 died due to ventricular fibrillation (red and black box).

**Figure 6 cells-09-00551-f006:**
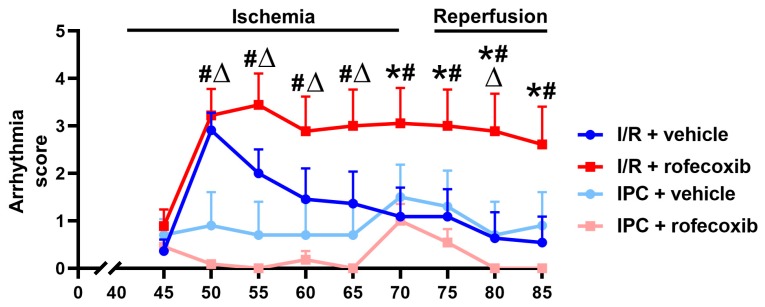
Arrhythmia scores declined gradually starting from the 50^th^ min. in the I/R+vehicle (ischemia/reperfusion) group but remained elevated in the I/R + rofecoxib group (* *p* < 0.05 I/R + vehicle vs. I/R + rofecoxib, *n* = 11–18). IPC (ischemic preconditioning) prevented initial increase of arrhythmia score (# *p* < 0.05 IPC + rofecoxib vs. I/R + rofecoxib, ∆*p* < 0.05 IPC + vehicle vs. I/R + rofecoxib, *n* = 10–11).

**Figure 7 cells-09-00551-f007:**
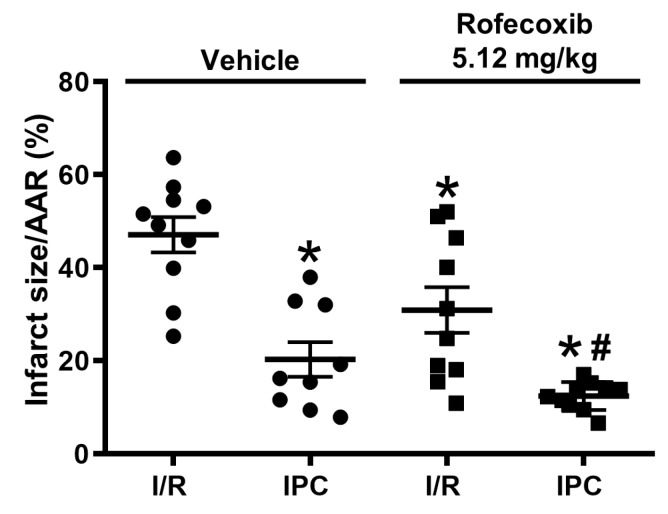
Chronic rofecoxib treatment reduced infarct size and did not interfere with cardioprotection by ischemic preconditioning. (* *p* < 0.05 vs. I/R + vehicle, # *p* < 0.05 vs. I/R + rofecoxib, *n* = 9–10).

**Figure 8 cells-09-00551-f008:**
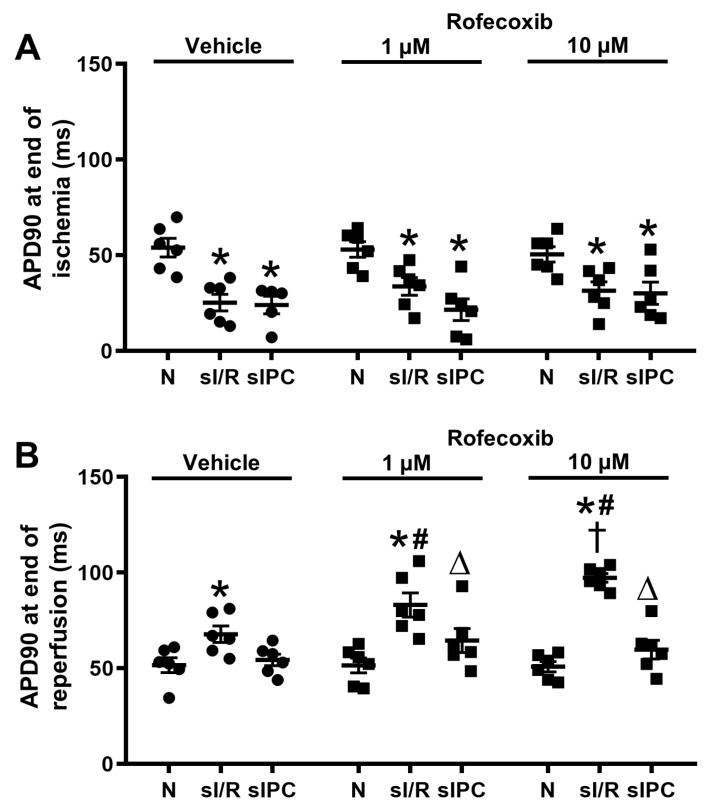
(**A**) Action potential duration at 90% repolarization (APD_90_) decreased by the end of 30 min. simulated ischemia in the simulated ischemia/reperfusion groups (sI/R) and simulated ischemic preconditioning groups (sIPC) as compared to the normoxia (N) group. (**B**) Rofecoxib increased the APD_90_ in adult rat isolated papillary muscles at the end of reperfusion and this effect was reversed by sIPC (**p* < 0.05 vs. corresponding normoxia group, #*p* < 0.05 vs. sI/R + vehicle,†*p* < 0.05 vs. sI/R + 1 µM rofecoxib, ∆*p* < 0.05 vs. corresponding sI/R group, *n* = 5–6).

**Figure 9 cells-09-00551-f009:**
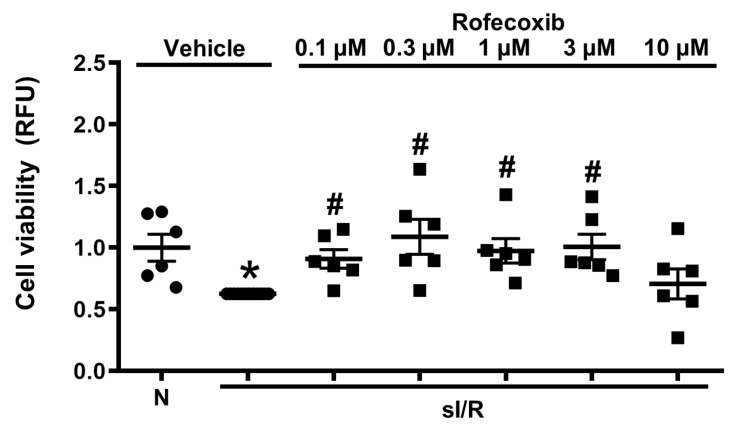
Rofecoxib increased cell viability in isolated rat cardiac myocytes exposed to simulated ischemia/reperfusion (sI/R). Normoxia (N) + vehicle group was set to 1 relative fluorescence units (RFU) arbitrary unit and all of the data were normalized to the averaged sI/R group (* *p* < 0.05 vs. Normoxia+vehicle, # *p* < 0.05 vs. sI/R + vehicle, *n* = 6). RFU-arbitrary unit: Relative fluorescence unit.

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
