# Peer review of "Hidden Cardiotoxicity of Rofecoxib Can be Revealed in Experimental Models of Ischemia/Reperfusion"

_cells, 2020, doi:10.3390/cells9030551_

Round 1

Reviewer 1 Report

Dear authors,

I believe that this topic about the rofecoxib cardiotoxcity in heart-diseased patients is very interesting. The manuscript is well written in all the sections. The topic is well presented in the introduction and the discussion fits with the results. The work is very articulated, comprising experiments in vivo, ex vivo and in vitro, to describe the cardiac effects of rofecoxib. 

I have only few corrections to suggest:

line 198: add a point after (sI/R groups);  line 199: substitute "next" with "following"; in the supplementary material, in the last line an highlighting is present.

Author Response

We thank the reviewer for their comments. We revised the manuscript accordingly. 

Remark 1: line 198: add a point after (sI/R groups);  line 199: substitute "next" with "following"; in the supplementary material, in the last line an highlighting is present.

Answer: Please see the changes in the revised form of the manuscript highlighted by “track changes” on page 5. The highlighting in the supplementary material was not present in our uploaded version. We are going to reupload the supplementary material.

Reviewer 2 Report

Reviewed manuscript from authors Brenner et al. deals with unrevealed cardiotoxic potential of drugs during their development. It is important preclinical and clinical issue. I agree with the authors' idea that exclusion of patients/animals with preexisting CVS morbidities from clinical/preclinical studies is not good idea in general and "the lost of valuable safety infromation" could be dangerous to patients and costly issue for R&D.

Manuscript is descriptive, methodological based, however a lack of deeper insight to molecular mechanisms of rofecoxib effects is a drawback of this paper. Although it is overall well written (with good and clear English, only few typos) and clearly describes backgrounds and methodology used, and discussion is comprehensive, there are some aspects that need to be clarify before publishing:

     in Abstact - there is a apparent discrepancy for readers regarding cardioprotective side of rofecoxib. It is presented as a bad guy, also in the Introduction. it shluld be mentioned before that there are also other studies showing this possible bright side of rofecoxib.

     in Materials and Methods section - please state exactly number of animals in each group for particular experiment.

     Fig. 5 - I'd find interesting to mention and distinguished in arrythmia maps which particular animals died in rofecoxib treated groups

     ad 3.3, l. 302 - "34% reduction in infarct size" - did you mean I/R veh vs. IPC rofecoxib? You changed the way of interpretation compared to previous figures. I'd prefere to interprete the decrease in each group separately (i.e. I/R veh vs I/R rofe, and IPC veh vs IPC rofe)  

     ad 3.4 & Fig. 8. - please amend suffix A or B when describing rofecoxib dose-dependent changes in ADP... it will more easy for reading as you have started to describe Fig 8B prior to 8A.

     ad 3.5 - isolated adult cardiac myocytes - how many animals were in each group? It seems to me that there was only one rat per group and isolated cell swere measured 6 times (l. 250, p.7: six separate technical replicates). Furthermore, I've never seen such a uniform data from isolated adult cardiac cells as in vehicle-treated sI/R group, do the authors have any explanation for this?

     p.10, l.341 - "thereby supporting the in vivo data on cellular level."I am confused, which in vivo data's showed that rofecoxib in I/R setting is cardioprotective??? In IPC group with rofecoxib, there was a decrease in infarct size. Could the authors verify the same "protective" effect during IPC in rats using single simulated I/R on cells???

Author Response

We thank the reviewer for their comments. We revised the manuscript accordingly. 

Remark 1: in Abstact - there is a apparent discrepancy for readers regarding cardioprotective side of rofecoxib. It is presented as a bad guy, also in the Introduction. it shluld be mentioned before that there are also other studies showing this possible bright side of rofecoxib.

Answer 1: We further clarified in the abstract that while rofecoxib showed hidden cardiotoxic effect manifested as a proarrthythmic effect during I/R, interestingly it decreased infarct size on page 1 as follows:

“Interestingly, while showing hidden cardiotoxicity manifested as a proarrhythmic effect during I/R, rofecoxib decreased infarct size and increased survival of adult rat cardiac myocytes subjected to simulated I/R injury.”

Ramark 2: in Materials and Methods section - please state exactly number of animals in each group for particular experiment.

Answer 2: Answer: According to the request of the reviewer, we added the group size of each groups during the study and described assignment of animals to the different groups to reach a comparable number surviving animals in each group. We amended the methods section accordingly as follows on page 3 and 4:

“Altogether 62 animals were included in the in vivo experiments. In order to achieve comparable number of surviving animals in each group, based on our preliminary observations 30% more animals were assigned to the rofecoxib-treated group (n=35) than to the vehicle-treated group (n=27). Rofecoxib- and vehicle-treated animals were then subjected to I/R with or without IPC using directed randomization during the study to assign more animals to the higher mortality groups: I/R+vehicle group (n=11), I/R+rofecoxib group (n=18), IPC+vehicle group (n=16) and IPC+rofecoxib group (n=17). I/R was induced by 30 min LAD occlusion and IPC was elicited by 3 cycles of brief 5-min LAD occlusion and 5-min reperfusion before I/R.”

On page 5:

“The experimental design and study protocols are illustrated on Figure 2. Altogether 54 animals were included in the ex vivo experiments. Papillary muscles of 6 animals/group were superfused with oxygen–saturated HEPES-buffered Tyrode’s solution (normoxic solution) and were allowed to equilibrate for 60 min before baseline measurements were taken.”

On page 6:

“The experimental design and study protocols are illustrated on Figure 3. Altogether 12 animals were included in the in vitro experiments. To achieve the n=6 group size in all the different normoxic and sI/R groups, i.e. 6 separate series of cell isolation procedures were made using 2 hearts, 1 for the normoxic and 1 for the sI/R groups.”

Remark 3: Fig. 5 - I'd find interesting to mention and distinguished in arrythmia maps which particular animals died in rofecoxib treated groups

Answer 3: According to the request of the reviewer, we modified the figure 5 and amended the figure legends to clarify the arrhythmia map and death of animals on page 8 and 9 as follows:

“Figure 5. Arrhythmia maps showing the arrhythmias in the order of severity during 30 min ischemia and at the first 15 min of reperfusion. Each row represents arrhythmias of each animal. The different color boxes show 5-min periods. Animals died during the IPC (ischemic preconditioning) are not shown. In the I/R+rofecoxib group animals 1-7 died due to ventricular fibrillation (red and black box).”

Remark 4: ad 3.3, l. 302 - "34% reduction in infarct size" - did you mean I/R veh vs. IPC rofecoxib? You changed the way of interpretation compared to previous figures. I'd prefere to interprete the decrease in each group separately (i.e. I/R veh vs I/R rofe, and IPC veh vs IPC rofe) 

Answer 4: we deleted the confusing calculation and simplified the text on page 9 as follows:

“Rofecoxib reduced infarct size (I/R+rofecoxib) as compared to vehicle-treated (I/R+vehicle) group (Figure 7).”

Remark 5: ad 3.4 & Fig. 8. - please amend suffix A or B when describing rofecoxib dose-dependent changes in ADP... it will more easy for reading as you have started to describe Fig 8B prior to 8A.

Answer 5: For easier reading, we clarified references for Figures on page 10 and Supplementary figure as follows:

“Rofecoxib treatment did not change any of the investigated electrophysiological parameters, including APD90 (Figure 8A and 8B) and APD75 (Supplementary figure 1A and 1B) in normoxic conditions. As expected, the 30 min simulated ischemia significantly shortened APD90 (Figure 8A) and APD75 (Supplementary figure 1A) in all groups subjected to ischemia compared to the respective normoxic groups. Importantly, however, in the presence of sI/R rofecoxib dose-dependently increased APD90 (Figure 8B) and increased APD75 (Supplementary figure 1B) upon reperfusion following the 30 min simulated ischemia. In the sIPC group these effects of rofecoxib on APD were not seen during reperfusion (Figure 8B and Supplementary figure 1B).”

Remark 6: ad 3.5 - isolated adult cardiac myocytes - how many animals were in each group? It seems to me that there was only one rat per group and isolated cell swere measured 6 times (l. 250, p.7: six separate technical replicates). Furthermore, I've never seen such a uniform data from isolated adult cardiac cells as in vehicle-treated sI/R group, do the authors have any explanation for this?

Answer 6: As to the animal numbers in each group: see answer above on number of animals. As to the uniformity of the data: the reason for the uniformity of data is that the fluorescence signal intensities were averaged for each group from 4 wells, and normalized to the SI / R groups to enable one to compare the different series of experiments. Therefore, the averages of the SI / R groups in Figure 9 are the same.

We clarified this in the text on page 7 as follows:

“Six separate technical repeats were performed, average of 4 wells/group/repeats are presented on graph. Cell survival data are expressed as relative fluorescence units (RFU). Normoxia+vehicle group was set to 1 RFU arbitrary unit and all data were normalized to the averaged sI/R group.”

Remark 7: p.10, l.341 - "thereby supporting the in vivo data on cellular level."I am confused, which in vivo data's showed that rofecoxib in I/R setting is cardioprotective??? In IPC group with rofecoxib, there was a decrease in infarct size. Could the authors verify the same "protective" effect during IPC in rats using single simulated I/R on cells???

Answer 7: As to infarct size data we clarified the text on page 11.

“sI/R caused significant cell death (Figure 9) compared to normoxic control which was reversed by rofecoxib treatment at 0.1, 0.3, 1 and 3 µM concentration, respectively, thereby supporting the in vivo data showing the infarct size reduction by rofecoxib (Figure 7).”

Answer 7: As to IPC on cells: Indeed, it would be nice to amend the experiments with IPC in the cell culture experiments, however, in our hands, IPC cannot be induced in this system, mainly due to technical problems to induce rapid cycles of changes in oxygen tension.